# Spectral Methods for Supervised Topic Models

**Yining Wang**[†]      **Jun Zhu**[‡]
[†]Machine Learning Department, Carnegie Mellon University, `yiningwa@cs.cmu.edu`
[‡]Dept. of Comp. Sci. & Tech.; Tsinghua National TNList Lab; State Key Lab of Intell. Tech. & Sys.,
Tsinghua University, `dcszj@mail.tsinghua.edu.cn`

## Abstract

Supervised topic models simultaneously model the latent topic structure of large collections of documents and a response variable associated with each document. Existing inference methods are based on either variational approximation or Monte Carlo sampling. This paper presents a novel spectral decomposition algorithm to recover the parameters of supervised latent Dirichlet allocation (sLDA) models. The Spectral-sLDA algorithm is provably correct and computationally efficient. We prove a sample complexity bound and subsequently derive a sufficient condition for the identifiability of sLDA. Thorough experiments on a diverse range of synthetic and real-world datasets verify the theory and demonstrate the practical effectiveness of the algorithm.

## 1   Introduction

Topic modeling offers a suite of useful tools that automatically learn the latent semantic structure of a large collection of documents. Latent Dirichlet allocation (LDA) [9] represents one of the most popular topic models. The vanilla LDA is an unsupervised model built on input contents of documents. In many applications side information is available apart from raw contents, e.g., user-provided rating scores of an online review text. Such side signal usually provides additional information to reveal the underlying structures of the documents in study. There have been extensive studies on developing topic models that incorporate various side information, e.g., by treating it as supervision. Some representative models are supervised LDA (sLDA) [8] that captures a real-valued regression response for each document, multiclass sLDA [21] that learns with discrete classification responses, discriminative LDA (DiscLDA) [14] that incorporates classification response via discriminative linear transformations on topic mixing vectors, and MedLDA [22, 23] that employs a max-margin criterion to learn discriminative latent topic representations.

Topic models are typically learned by finding maximum likelihood estimates (MLE) through local search or sampling methods [12, 18, 19], which may suffer from local optima. Much recent progress has been made on developing spectral decomposition [1, 2, 3] and nonnegative matrix factorization (NMF) [4, 5, 6, 7] methods to infer latent topic-word distributions. Instead of finding MLE estimates, which is a known NP-hard problem [6], these methods assume that the documents are i.i.d. sampled from a topic model, and attempt to recover the underlying model parameters. Compared to local search and sampling algorithms, these methods enjoy the advantage of being provably effective. In fact, sample complexity bounds have been proved to show that given a sufficiently large collection of documents, these algorithms can recover the model parameters accurately with a high probability.

Although spectral decomposition (as well as NMF) methods have achieved increasing success in recovering latent variable models, their applicability is quite limited. For example, previous work has mainly focused on unsupervised latent variable models, leaving the broad family of supervised models (e.g., sLDA) largely unexplored. The only exception is [10] which presents a spectral method for mixtures of regression models, quite different from sLDA. Such ignorance is not a coincidence as supervised models impose new technical challenges. For instance, a direct application of previous

techniques [1, 2] on sLDA cannot handle regression models with duplicate entries. In addition, the sample complexity bound gets much worse if we try to match entries in regression models with their corresponding topic vectors. On the practical side, few quantitative experimental results (if any at all) are available for spectral decomposition based methods on LDA models.

In this paper, we extend the applicability of spectral learning methods by presenting a novel spectral decomposition algorithm to recover the parameters of sLDA models from empirical low-order moments estimated from the data. We provide a sample complexity bound and analyze the identifiability conditions. A key step in our algorithm is a power update step that recovers the regression model in sLDA. The method uses a newly designed empirical moment to recover regression model entries directly from the data and reconstructed topic distributions. It is free from making any constraints on the underlying regression model, and does not increase the sample complexity much. We also provide thorough experiments on both synthetic and real-world datasets to demonstrate the practical effectiveness of our proposed algorithm. By combining our spectral recovery algorithm with a Gibbs sampling procedure, we showed superior performance in terms of language modeling, prediction accuracy and running time compared to traditional inference algorithms.

## 2 Preliminaries

We first overview the basics of sLDA, orthogonal tensor decomposition and the notations to be used.

### 2.1 Supervised LDA

Latent Dirichlet allocation (LDA) [9] is a generative model for topic modeling of text documents. It assumes $k$ different *topics* with topic-word distributions $\boldsymbol{\mu}_1, \cdots, \boldsymbol{\mu}_k \in \Delta^{V-1}$, where $V$ is the vocabulary size and $\Delta^{V-1}$ denotes the probability simplex of a $V$-dimensional random vector. For a document, LDA models a *topic mixing vector* $\boldsymbol{h} \in \Delta^{k-1}$ as a probability distribution over the $k$ topics. A conjugate Dirichlet prior with parameter $\boldsymbol{\alpha}$ is imposed on the topic mixing vectors. A bag-of-word model is then adopted, which generates each word in the document based on $\boldsymbol{h}$ and the topic-word vectors $\boldsymbol{\mu}$. Supervised latent Dirichlet allocation (sLDA) [8] incorporates an extra response variable $y \in \mathbb{R}$ for each document. The response variable is modeled by a linear regression model $\boldsymbol{\eta} \in \mathbb{R}^k$ on either the topic mixing vector $\boldsymbol{h}$ or the averaging topic assignment vector $\bar{\boldsymbol{z}}$, where $\bar{z}_i = \frac{1}{m} \sum_j 1_{[z_j=i]}$ with $m$ the number of words in a document. The noise is assumed to be Gaussian with zero mean and $\sigma^2$ variance.

Fig. 1 shows the graph structure of two sLDA variants mentioned above. Although previous work has mainly focused on model (b) which is convenient for Gibbs sampling and variational inference, we consider model (a) because it will considerably simplify our spectral algorithm and analysis. One may assume that whenever a document is not too short, the empirical distribution of its word topic assignments should be close to the document's topic mixing vector. Such a scheme was adopted to learn sparse topic coding models [24], and has demonstrated promising results in practice.

### 2.2 High-order tensor product and orthogonal tensor decomposition

A real *p-th order tensor* $A \in \bigotimes_{i=1}^p \mathbb{R}^{n_i}$ belongs to the tensor product of Euclidean spaces $\mathbb{R}^{n_i}$. Generally we assume $n_1 = n_2 = \cdots = n_p = n$, and we can identify each coordinate of $A$ by a $p$-tuple $(i_1, \cdots, i_p)$, where $i_1, \cdots, i_p \in [n]$. For instance, a $p$-th order tensor is a vector when $p = 1$ and a matrix when $p = 2$. We can also consider a $p$-th order tensor $A$ as a multilinear mapping. For $A \in \bigotimes^p \mathbb{R}^n$ and matrices $X_1, \cdots, X_p \in \mathbb{R}^{n \times m}$, the mapping $A(X_1, \cdots, X_p)$ is a $p$-th order tensor in $\bigotimes^p \mathbb{R}^m$, with $[A(X_1, \cdots, X_p)]_{i_1, \cdots, i_p} \triangleq \sum_{j_1, \cdots, j_p \in [n]} A_{j_1, \cdots, j_p} [X_1]_{j_1, i_1} [X_2]_{j_2, i_2} \cdots [X_p]_{j_p, i_p}$. Consider some concrete examples of such a multilinear mapping. When $A, X_1, X_2$ are matrices, we have $A(X_1, X_2) = X_1^\top A X_2$. Similarly, when $A$ is a matrix and $x$ is a vector, $A(I, x) = Ax$.

An orthogonal tensor decomposition of a tensor $A \in \bigotimes^p \mathbb{R}^n$ is a collection of orthonormal vectors $\{\boldsymbol{v}_i\}_{i=1}^k$ and scalars $\{\lambda_i\}_{i=1}^k$ such that $A = \sum_{i=1}^k \lambda_i \boldsymbol{v}_i^{\otimes p}$. Without loss of generality, we assume $\lambda_i$ are nonnegative when $p$ is odd. Although orthogonal tensor decomposition in the matrix case can be done efficiently by singular value decomposition (SVD), it has several delicate issues in higher order tensor spaces [2]. For instance, tensors may not have unique decompositions, and an orthogonal decomposition may not exist for every symmetric tensor [2]. Such issues are further complicated when only noisy estimates of the desired tensors are available. For these reasons, we need more advanced techniques to handle high-order tensors. In this paper, we will apply the *robust*

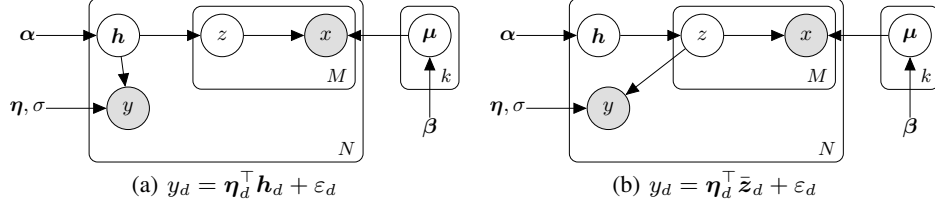

(a) $y_d = \boldsymbol{\eta}_d^\top \boldsymbol{h}_d + \varepsilon_d$            (b) $y_d = \boldsymbol{\eta}_d^\top \bar{\boldsymbol{z}}_d + \varepsilon_d$

Figure 1: Plate notations for two variants of sLDA

*tensor power method* [2] to recover robust eigenvalues and eigenvectors of an (estimated) third-order tensor. The algorithm recovers eigenvalues and eigenvectors up to an absolute error $\varepsilon$, while running in polynomial time with respect to the tensor dimension and $\log(1/\varepsilon)$. Further details and analysis of the robust tensor power method are presented in Appendix A.2 and [2].

### 2.3 Notations

Throughout, we use $\boldsymbol{v}^{\otimes p} \triangleq \boldsymbol{v} \otimes \boldsymbol{v} \otimes \cdots \otimes \boldsymbol{v}$ to denote the $p$-th order tensor generated by a vector $\boldsymbol{v}$. We use $\|\boldsymbol{v}\| = \sqrt{\sum_i v_i^2}$ to denote the Euclidean norm of a vector $\boldsymbol{v}$, $\|M\|$ to denote the spectral norm of a matrix $M$ and $\|T\|$ to denote the operator norm of a high-order tensor. $\|M\|_F = \sqrt{\sum_{i,j} M_{ij}^2}$ denotes the Frobenious norm of a matrix. We use an indicator vector $\boldsymbol{x} \in \mathbb{R}^V$ to represent a word in a document, e.g., for the $i$-th word in the vocabulary, $x_i = 1$ and $x_j = 0$ for all $j \neq i$. We also use $O \triangleq (\boldsymbol{\mu}_1, \boldsymbol{\mu}_2, \cdots, \boldsymbol{\mu}_k) \in \mathbb{R}^{V \times k}$ to denote the topic distribution matrix, and $\widetilde{O} \triangleq (\widetilde{\boldsymbol{\mu}}_1, \widetilde{\boldsymbol{\mu}}_2, \cdots, \widetilde{\boldsymbol{\mu}}_K)$ to denote the canonical version of $O$, where $\widetilde{\boldsymbol{\mu}}_i = \sqrt{\frac{\alpha_i}{\alpha_0(\alpha_0+1)}} \boldsymbol{\mu}$ with $\alpha_0 = \sum_{i=1}^k \alpha_i$.

## 3 Spectral Parameter Recovery

We now present a novel spectral parameter recovery algorithm for sLDA. The algorithm consists of two key components—the orthogonal tensor decomposition of observable moments to recover the topic distribution matrix $O$ and a power update method to recover the linear regression model $\boldsymbol{\eta}$. We elaborate on these techniques and a rigorous theoretical analysis in the following sections.

### 3.1 Moments of observable variables

Our spectral decomposition methods recover the topic distribution matrix $O$ and the linear regression model $\boldsymbol{\eta}$ by manipulating moments of observable variables. In Definition 1, we define a list of moments on random variables from the underlying sLDA model.

**Definition 1.** *We define the following moments of observable variables:*

$$M_1 = \mathbb{E}[\boldsymbol{x}_1], \quad M_2 = \mathbb{E}[\boldsymbol{x}_1 \otimes \boldsymbol{x}_2] - \frac{\alpha_0}{\alpha_0 + 1} M_1 \otimes M_1, \tag{1}$$

$$M_3 = \mathbb{E}[\boldsymbol{x}_1 \otimes \boldsymbol{x}_2 \otimes \boldsymbol{x}_3] - \frac{\alpha_0}{\alpha_0 + 2} \left( \mathbb{E}[\boldsymbol{x}_1 \otimes \boldsymbol{x}_2 \otimes M_1] + \mathbb{E}[\boldsymbol{x}_1 \otimes M_1 \otimes \boldsymbol{x}_2] + \mathbb{E}[M_1 \otimes \boldsymbol{x}_1 \otimes \boldsymbol{x}_2] \right)$$

$$+ \frac{2\alpha_0^2}{(\alpha_0 + 1)(\alpha_0 + 2)} M_1 \otimes M_1 \otimes M_1, \tag{2}$$

$$M_y = \mathbb{E}[y \boldsymbol{x}_1 \otimes \boldsymbol{x}_2] - \frac{\alpha_0}{\alpha_0 + 2} \left( \mathbb{E}[y] \mathbb{E}[\boldsymbol{x}_1 \otimes \boldsymbol{x}_2] + \mathbb{E}[\boldsymbol{x}_1] \otimes \mathbb{E}[y \boldsymbol{x}_2] + \mathbb{E}[y \boldsymbol{x}_1] \otimes \mathbb{E}[\boldsymbol{x}_2] \right)$$

$$+ \frac{2\alpha_0^2}{(\alpha_0 + 1)(\alpha_0 + 2)} \mathbb{E}[y] M_1 \otimes M_1. \tag{3}$$

Note that the moments $M_1$, $M_2$ and $M_3$ were also defined and used in previous work [1, 2] for the parameter recovery for LDA models. For the sLDA model, we need to define a new moment $M_y$ in order to recover the linear regression model $\boldsymbol{\eta}$. The moments are based on observable variables in the sense that they can be estimated from i.i.d. sampled documents. For instance, $M_1$ can be estimated by computing the empirical distribution of all words, and $M_2$ can be estimated using $M_1$ and word co-occurrence frequencies. Though the moments in the above forms look complicated, we can apply elementary calculations based on the conditional independence structure of sLDA to significantly simplify them and more importantly to get them connected with the model parameters to be recovered, as summarized in Proposition 1. The proof is deferred to Appendix B.

**Proposition 1.** *The moments can be expressed using the model parameters as:*

$$M_2 = \frac{1}{\alpha_0(\alpha_0+1)} \sum_{i=1}^{k} \alpha_i \boldsymbol{\mu}_i \otimes \boldsymbol{\mu}_i, \quad M_3 = \frac{2}{\alpha_0(\alpha_0+1)(\alpha_0+2)} \sum_{i=1}^{k} \alpha_i \boldsymbol{\mu}_i \otimes \boldsymbol{\mu}_i \otimes \boldsymbol{\mu}_i, \tag{4}$$

$$M_y = \frac{2}{\alpha_0(\alpha_0+1)(\alpha_0+2)} \sum_{i=1}^{k} \alpha_i \eta_i \boldsymbol{\mu}_i \otimes \boldsymbol{\mu}_i. \tag{5}$$

## 3.2 Simultaneous diagonalization

Proposition 1 shows that the moments in Definition 1 are all the weighted sums of tensor products of $\{\boldsymbol{\mu}_i\}_{i=1}^{k}$ from the underlying sLDA model. One idea to reconstruct $\{\boldsymbol{\mu}_i\}_{i=1}^{k}$ is to perform *simultaneous diagonalization* on tensors of different orders. The idea has been used in a number of recent developments of spectral methods for latent variable models [1, 2, 10]. Specifically, we first whiten the second-order tensor $M_2$ by finding a matrix $W \in \mathbb{R}^{V \times k}$ such that $W^\top M_2 W = I_k$. This whitening procedure is possible whenever the topic distribution vectors $\{\boldsymbol{\mu}_i\}_{i=1}^{k}$ are linearly independent (and hence $M_2$ has rank $k$). The whitening procedure and the linear independence assumption also imply that $\{W\boldsymbol{\mu}_i\}_{i=1}^{k}$ are orthogonal vectors (see Appendix A.2 for details), and can be subsequently recovered by performing an orthogonal tensor decomposition on the simultaneously whitened third-order tensor $M_3(W, W, W)$. Finally, by multiplying the pseudo-inverse of the whitening matrix $W^+$ we obtain the topic distribution vectors $\{\boldsymbol{\mu}_i\}_{i=1}^{k}$.

It should be noted that Jennrich's algorithm [13, 15, 17] could recover $\{\boldsymbol{\mu}_i\}_{i=1}^{k}$ directly from the 3-rd order tensor $M_3$ alone when $\{\boldsymbol{\mu}_i\}_{i=1}^{k}$ is linearly independent. However, we still adopt the above simultaneous diagonalization framework because the intermediate vectors $\{W\boldsymbol{\mu}_i\}_{i=1}^{k}$ play a vital role in the recovery procedure of the linear regression model $\boldsymbol{\eta}$.

## 3.3 The power update method

Although the linear regression model $\boldsymbol{\eta}$ can be recovered in a similar manner by performing simultaneous diagonalization on $M_2$ and $M_y$, such a method has several disadvantages, thereby calling for novel solutions. First, after obtaining entry values $\{\eta_i\}_{i=1}^{k}$ we need to match them to the topic distributions $\{\boldsymbol{\mu}_i\}_{i=1}^{k}$ previously recovered. This can be easily done when we have access to the true moments, but becomes difficult when only estimates of observable tensors are available because the estimated moments may not share the same singular vectors due to sampling noise. A more serious problem is that when $\boldsymbol{\eta}$ has duplicate entries the orthogonal decomposition of $M_y$ is no longer unique. Though a randomized strategy similar to the one used in [1] might solve the problem, it could substantially increase the sample complexity [2] and render the algorithm impractical.

We develop a power update method to resolve the above difficulties. Specifically, after obtaining the whitened (orthonormal) vectors $\{\boldsymbol{v}_i\} \triangleq c_i \cdot W\boldsymbol{\mu}_i$ [1] we recover the entry $\eta_i$ of the linear regression model directly by computing a power update $\boldsymbol{v}_i^\top M_y(W, W)\boldsymbol{v}_i$. In this way, the matching problem is automatically solved because we know what topic distribution vector $\boldsymbol{\mu}_i$ is used when recovering $\eta_i$. Furthermore, the singular values (corresponding to the entries of $\boldsymbol{\eta}$) do not need to be distinct because we are not using any unique SVD properties of $M_y(W, W)$. As a result, our proposed algorithm works for any linear model $\boldsymbol{\eta}$.

## 3.4 Parameter recovery algorithm

An outline of our parameter recovery algorithm for sLDA (Spectral-sLDA) is given in Alg. 1. First, empirical estimates of the observable moments in Definition 1 are computed from the given documents. The simultaneous diagonalization method is then used to reconstruct the topic distribution matrix $O$ and its prior parameter $\boldsymbol{\alpha}$. After obtaining $O = (\boldsymbol{\mu}_1, \cdots, \boldsymbol{\mu}_k)$, we use the power update method introduced in the previous section to recover the linear regression model $\boldsymbol{\eta}$.

Alg. 1 admits three hyper-parameters $\alpha_0$, $L$ and $T$. $\alpha_0$ is defined as the sum of all entries in the prior parameter $\boldsymbol{\alpha}$. Following the conventions in [1, 2], we assume that $\alpha_0$ is known a priori and use this value to perform parameter estimation. It should be noted that this is a mild assumption, as in practice usually a homogeneous vector $\boldsymbol{\alpha}$ is assumed and the entire vector is known [20]. The $L$ and $T$ parameters are used to control the number of iterations in the robust tensor power method. In general, the robust tensor power method runs in $O(k^3 LT)$ time. To ensure sufficient recovery accuracy,

**Algorithm 1** spectral parameter recovery algorithm for sLDA. Input parameters: $\alpha_0, L, T$.

1: Compute empirical moments and obtain $\widehat{M_2}, \widehat{M_3}$ and $\widehat{M_y}$.
2: Find $\widehat{W} \in \mathbb{R}^{n \times k}$ such that $\widehat{M_2}(\widehat{W}, \widehat{W}) = I_k$.
3: Find robust eigenvalues and eigenvectors $(\widehat{\lambda}_i, \widehat{\boldsymbol{v}}_i)$ of $\widehat{M_3}(\widehat{W}, \widehat{W}, \widehat{W})$ using the robust tensor power method [2] with parameters $L$ and $T$.
4: Recover prior parameters: $\widehat{\alpha}_i \leftarrow \frac{4\alpha_0(\alpha_0+1)}{(\alpha_0+2)^2\widehat{\lambda}_i^2}$.
5: Recover topic distributions: $\widehat{\boldsymbol{\mu}}_i \leftarrow \frac{\alpha_0+2}{2}\widehat{\lambda}_i(\widehat{W}^+)^\top \widehat{\boldsymbol{v}}_i$.
6: Recover the linear regression model: $\widehat{\eta}_i \leftarrow \frac{\alpha_0+2}{2}\widehat{\boldsymbol{v}}_i^\top \widehat{M_y}(\widehat{W},\widehat{W})\widehat{\boldsymbol{v}}_i$.
7: **Output:** $\widehat{\boldsymbol{\eta}}$, $\widehat{\boldsymbol{\alpha}}$ and $\{\widehat{\boldsymbol{\mu}}_i\}_{i=1}^k$.

---

$L$ should be at least a linear function of $k$ and $T$ should be set as $T = \Omega(\log(k) + \log\log(\lambda_{\max}/\varepsilon))$, where $\lambda_{\max} = \frac{2}{\alpha_0+2}\sqrt{\frac{\alpha_0(\alpha_0+1)}{\alpha_{\min}}}$ and $\varepsilon$ is an error tolerance parameter. Appendix A.2 and [2] provide a deeper analysis into the choice of $L$ and $T$ parameters.

### 3.5 Speeding up moment computation

In Alg. 1, a straightforward computation of the third-order tensor $\widehat{M_3}$ requires $O(NM^3)$ time and $O(V^3)$ storage, where $N$ is corpus size and $M$ is the number of words per document. Such time and space complexities are clearly prohibitive for real applications, where the vocabulary usually contains tens of thousands of terms. However, we can employ a trick similar as in [11] to speed up the moment computation. We first note that only the whitened tensor $\widehat{M_3}(\widehat{W}, \widehat{W}, \widehat{W})$ is needed in our algorithm, which only takes $O(k^3)$ storage. Another observation is that the most difficult term in $\widehat{M_3}$ can be written as $\sum_{i=1}^r c_i \boldsymbol{u}_{i,1} \otimes \boldsymbol{u}_{i,2} \otimes \boldsymbol{u}_{i,3}$, where $r$ is proportional to $N$ and $\boldsymbol{u}_{i,\cdot}$ contains at most $M$ non-zero entries. This allows us to compute $\widehat{M_3}(\widehat{W}, \widehat{W}, \widehat{W})$ in $O(NMk)$ time by computing $\sum_{i=1}^r c_i(W^\top \boldsymbol{u}_{i,1}) \otimes (W^\top \boldsymbol{u}_{i,2}) \otimes (W^\top \boldsymbol{u}_{i,3})$. Appendix B.2 provides more details about this speed-up trick. The overall time complexity is $O(NM(M+k^2) + V^2 + k^3LT)$ and the space complexity is $O(V^2 + k^3)$.

## 4 Sample Complexity Analysis

We now analyze the sample complexity of Alg. 1 in order to achieve $\varepsilon$-error with a high probability. For clarity, we focus on presenting the main results, while deferring the proof details to Appendix A, including the proofs of important lemmas that are needed for the main theorem.

**Theorem 1.** *Let $\sigma_1(\widetilde{O})$ and $\sigma_k(\widetilde{O})$ be the largest and the smallest singular values of the canonical topic distribution matrix $\widetilde{O}$. Define $\lambda_{\min} \triangleq \frac{2}{\alpha_0+2}\sqrt{\frac{\alpha_0(\alpha_0+1)}{\alpha_{\max}}}$ and $\lambda_{\max} \triangleq \frac{2}{\alpha_0+2}\sqrt{\frac{\alpha_0(\alpha_0+1)}{\alpha_{\min}}}$ with $\alpha_{\max}$ and $\alpha_{\min}$ the largest and the smallest entries of $\boldsymbol{\alpha}$. Suppose $\widehat{\boldsymbol{\mu}}$, $\widehat{\boldsymbol{\alpha}}$ and $\widehat{\boldsymbol{\eta}}$ are the outputs of Algorithm 1, and $L$ is at least a linear function of $k$. Fix $\delta \in (0,1)$. For any small error-tolerance parameter $\varepsilon > 0$, if Algorithm 1 is run with parameter $T = \Omega(\log(k) + \log\log(\lambda_{\max}/\varepsilon))$ on $N$ i.i.d. sampled documents (each containing at least 3 words) with $N \geq \max(n_1, n_2, n_3)$, where*

$$n_1 = C_1 \cdot \left(1 + \sqrt{\log(6/\delta)}\right)^2 \cdot \frac{\alpha_0^2(\alpha_0+1)^2}{\alpha_{\min}}, \quad n_3 = C_3 \cdot \frac{(1+\sqrt{\log(9/\delta)})^2}{\sigma_k(\widetilde{O})^{10}} \cdot \max\left(\frac{1}{\varepsilon^2}, \frac{k^2}{\lambda_{\min}^2}\right),$$

$$n_2 = C_2 \cdot \frac{(1+\sqrt{\log(15/\delta)})^2}{\varepsilon^2\sigma_k(\widetilde{O})^4} \cdot \max\left((\|\boldsymbol{\eta}\| + \Phi^{-1}(\delta/60\sigma))^2, \alpha_{\max}^2\sigma_1(\widetilde{O})^2\right),$$

*and $C_1, C_2$ and $C_3$ are universal constants, then with probability at least $1 - \delta$, there exists a permutation $\pi : [k] \to [k]$ such that for every topic $i$, the following holds:*

  *1. $|\alpha_i - \widehat{\alpha}_{\pi(i)}| \leq \frac{4\alpha_0(\alpha_0+1)(\lambda_{\max}+5\varepsilon)}{(\alpha_0+2)^2\lambda_{\min}^2(\lambda_{\min}-5\varepsilon)^2} \cdot 5\varepsilon$, if $\lambda_{\min} > 5\varepsilon$;*

  *2. $\|\boldsymbol{\mu}_i - \widehat{\boldsymbol{\mu}}_{\pi(i)}\| \leq \left(3\sigma_1(\widetilde{O})\left(\frac{8\alpha_{\max}}{\lambda_{\min}} + \frac{5(\alpha_0+2)}{2}\right) + 1\right)\varepsilon$;*

  *3. $|\eta_i - \widehat{\eta}_{\pi(i)}| \leq \left(\frac{\|\boldsymbol{\eta}\|}{\lambda_{\min}} + (\alpha_0+2)\right)\varepsilon$.*

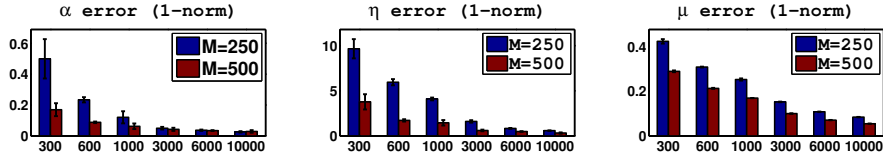

Figure 2: Reconstruction errors of Alg. 1. $X$ axis denotes the training size. Error bars denote the standard deviations measured on 3 independent trials under each setting.

In brevity, the proof is based on matrix perturbation lemmas (see Appendix A.1) and analysis to the orthogonal tensor decomposition methods (including SVD and robust tensor power method) performed on inaccurate tensor estimations (see Appendix A.2). The sample complexity lower bound consists of three terms, from $n_1$ to $n_3$. The $n_3$ term comes from the sample complexity bound for the robust tensor power method [2]; the $(\|\boldsymbol{\eta}\| + \Phi^{-1}(\delta/60\sigma))^2$ term in $n_2$ characterizes the recovery accuracy for the linear regression model $\boldsymbol{\eta}$, and the $\alpha_{\max}^2 \sigma_1(\widetilde{O})^2$ term arises when we try to recover the topic distribution vectors $\boldsymbol{\mu}$; finally, the term $n_1$ is required so that some technical conditions are met. The $n_1$ term does not depend on either $k$ or $\sigma_k(\widetilde{O})$, and could be largely neglected in practice.

An important implication of Theorem 1 is that it provides a sufficient condition for a supervised LDA model to be identifiable, as shown in Remark 1. To some extent, Remark 1 is the best identifiability result possible under our inference framework, because it makes no restriction on the linear regression model $\boldsymbol{\eta}$, and the linear independence assumption is unavoidable without making further assumptions on the topic distribution matrix $O$.

**Remark 1.** *Given a sufficiently large number of i.i.d. sampled documents with at least 3 words per document, a supervised LDA model $\mathcal{M} = (\boldsymbol{\alpha}, \boldsymbol{\mu}, \boldsymbol{\eta})$ is identifiable if $\alpha_0 = \sum_{i=1}^{k} \alpha_i$ is known and $\{\boldsymbol{\mu}_i\}_{i=1}^{k}$ are linearly independent.*

We also make remarks on indirected quantities appeared in Theorem 1 (e.g., $\sigma_k(\widetilde{O})$) and a simplified sample complexity bound for some special cases. They can be found in Appendix A.4.

## 5   Experiments

### 5.1   Datasets description and Algorithm implementation details

We perform experiments on both synthetic and real-world datasets. The synthetic data are generated in a similar manner as in [22], with a fixed vocabulary of size $V = 500$. We generate the topic distribution matrix $O$ by first sampling each entry from a uniform distribution and then normalize every column of $O$. The linear regression model $\boldsymbol{\eta}$ is sampled from a standard Gaussian distribution. The prior parameter $\boldsymbol{\alpha}$ is assumed to be homogeneous, i.e., $\boldsymbol{\alpha} = (1/k, \cdots, 1/k)$. Documents and response variables are then generated from the sLDA model specified in Sec. 2.1.

For real-world data, we use the large-scale dataset built on Amazon movie reviews [16] to demonstrate the practical effectiveness of our algorithm. The dataset contains 7,911,684 movie reviews written by 889,176 users from Aug 1997 to Oct 2012. Each movie review is accompanied with a score from 1 to 5 indicating how the user likes a particular movie. The median number of words per review is 101. A vocabulary with $V = 5,000$ terms is built by selecting high frequency words. We also pre-process the dataset by shifting the review scores so that they have zero mean.

Both Gibbs sampling for the sLDA model in Fig. 1 (b) and the proposed spectral recovery algorithm are implemented in C++. For our spectral algorithm, the hyperparameters $L$ and $T$ are set to 100, which is sufficiently large for all settings in our experiments. Since Alg. 1 can only recover the topic model itself, we use Gibbs sampling to iteratively sample topic mixing vectors $\boldsymbol{h}$ and topic assignments for each word $z$ in order to perform prediction on a held-out dataset.

### 5.2   Convergence of reconstructed model parameters

We demonstrate how the sLDA model reconstructed by Alg. 1 converges to the underlying true model when more observations are available. Fig. 2 presents the 1-norm reconstruction errors of $\boldsymbol{\alpha}$, $\boldsymbol{\eta}$ and $\boldsymbol{\mu}$. The number of topics $k$ is set to 20 and the number of words per document (i.e., $M$) is set

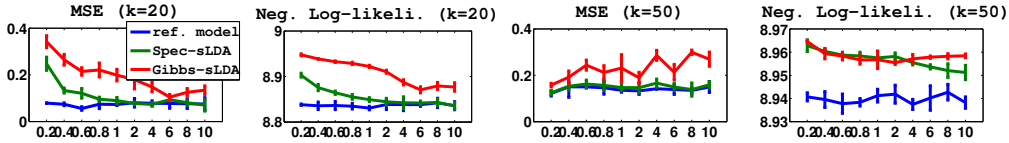

Figure 3: Mean square errors and negative per-word log-likelihood of Alg. 1 and Gibbs sLDA. Each document contains $M = 500$ words. The $X$ axis denotes the training size ($\times 10^3$).

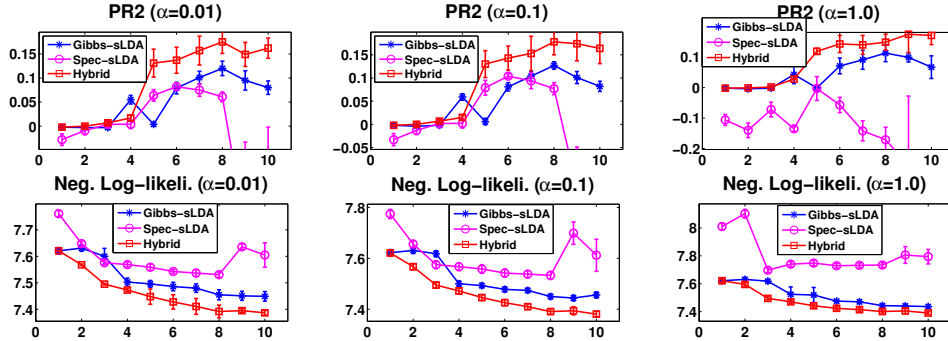

Figure 4: pR$^2$ scores and negative per-word log-likelihood. The $X$ axis indicates the number of topics. Error bars indicate the standard deviation of 5-fold cross-validation.

to 250 and 500. Since Spectral-sLDA can only recover topic distributions up to a permutation over $[k]$, a minimum weighted graph match was computed on $O$ and $\widehat{O}$ to find an optimal permutation.

Fig. 2 shows that the reconstruction errors for all the parameters go down rapidly as we obtain more documents. Furthermore, though Theorem 1 does not involve the number of words per document, the simulation results demonstrate a significant improvement when more words are observed in each document, which is a nice complement for the theoretical analysis.

### 5.3 Prediction accuracy and per-word likelihood

We compare the prediction accuracy and per-word likelihood of Spectral-sLDA and Gibbs-sLDA on both synthetic and real-world datasets. On the synthetic dataset, the regression error is measured by the mean square error (MSE), and the per-word log-likelihood is defined as $\log_2 p(w|\boldsymbol{h}, O) = \log_2 \sum_{k=1}^{K} p(w|z = k, O) p(z = k|\boldsymbol{h})$. The hyper-parameters used in our Gibbs sampling implementation are the same with the ones used to generate the datasets.

Fig. 3 shows that Spectral-sLDA consistently outperforms Gibbs-sLDA. Our algorithm also enjoys the advantage of being less variable, as indicated by the curve and error bars. Moreover, when the number of training documents is sufficiently large, the performance of the reconstructed model is very close to the underlying true model[2], which implies that Alg. 1 can correctly identify an sLDA model from its observations, therefore supporting our theory.

We also test both algorithms on the large-scale Amazon movie review dataset. The quality of the prediction is assessed with predictive R$^2$ (pR$^2$) [8], a normalized version of MSE, which is defined as $\text{pR}^2 \triangleq 1 - (\sum_i (y_i - \widehat{y}_i)^2)/(\sum_i (y_i - \bar{y})^2)$, where $\widehat{y}_i$ is the estimate, $y_i$ is the truth, and $\bar{y}$ is the average true value. We report the results under various settings of $\boldsymbol{\alpha}$ and $k$ in Fig. 4, with the $\sigma$ hyper-parameter of Gibbs-sLDA selected via cross-validation on a smaller subset of documents. Apart from Gibbs-sLDA and Spectral-sLDA, we also test the performance of a hybrid algorithm which performs Gibbs sampling using models reconstructed by Spectral-sLDA as initializations.

Fig. 4 shows that in general Spectral-sLDA does not perform as well as Gibbs sampling. One possible reason is that real-world datasets are not exact i.i.d. samples from an underlying sLDA model. However, a significant improvement can be observed when the Gibbs sampler is initialized with models reconstructed by Spectral-sLDA instead of random initializations. This is because Spectral-sLDA help avoid the local optimum problem of local search methods like Gibbs sampling. Similar improvements for spectral methods were also observed in previous papers [10].

Table 1: Training time of Gibbs-sLDA and Spectral-sLDA, measured in minutes. $k$ is the number of topics and $n$ is the number of documents used in training.

| $n(\times 10^4)$ | $k = 10$ | | | | | $k = 50$ | | | | |
|---|---|---|---|---|---|---|---|---|---|---|
| | 1 | 5 | 10 | 50 | 100 | 1 | 5 | 10 | 50 | 100 |
| Gibbs-sLDA | 0.6 | 3.0 | 6.0 | 30.5 | 61.1 | 2.9 | 14.3 | 28.2 | 145.4 | 281.8 |
| Spec-sLDA | 1.5 | 1.6 | 1.7 | 2.9 | 4.3 | 3.1 | 3.6 | 4.3 | 9.5 | 16.2 |

Table 2: Prediction accuracy and per-word log-likelihood of Gibbs-sLDA and the hybrid algorithm. The initialization solution is obtained by running Alg. 1 on a collection of 1 million documents, while $n$ is the number of documents used in Gibbs sampling. $k = 8$ topics are used.

| $\log_{10} n$ | predictive $R^2$ | | | | Negative per-word log-likelihood | | | |
|---|---|---|---|---|---|---|---|---|
| | 3 | 4 | 5 | 6 | 3 | 4 | 5 | 6 |
| Gibbs-sLDA | 0.00 | 0.04 | 0.11 | 0.14 | 7.72 | 7.55 | 7.45 | 7.42 |
| | (0.01) | (0.02) | (0.02) | (0.01) | (0.01) | (0.01) | (0.01) | (0.01) |
| Hybrid | 0.02 | 0.17 | 0.18 | 0.18 | 7.70 | 7.49 | 7.40 | 7.36 |
| | (0.01) | (0.03) | (0.03) | (0.03) | (0.01) | (0.02) | (0.01) | (0.01) |

Note that for $k > 8$ the performance of Spectral-sLDA significantly deteriorates. This phenomenon can be explained by the nature of Spectral-sLDA itself: one crucial step in Alg. 1 is to whiten the empirical moment $\widehat{M_2}$, which is only possible when the underlying topic matrix $O$ has full rank. For the Amazon movie review dataset, it is impossible to whiten $\widehat{M_2}$ when the underlying model contains more than 8 topics. This interesting observation shows that the Spectral-sLDA algorithm can be used for model selection to avoid overfitting by using too many topics.

## 5.4 Time efficiency

The proposed spectral recovery algorithm is very time efficient because it avoids time-consuming iterative steps in traditional inference and sampling methods. Furthermore, empirical moment computation, the most time-consuming part in Alg. 1, consists of only elementary operations and could be easily optimized. Table 1 compares the training time of Gibbs-sLDA and Spectral-sLDA and shows that our proposed algorithm is over 15 times faster than Gibbs sampling, especially for large document collections. Although both algorithms are implemented in a single-threading manner, Spectral-sLDA is very easy to parallelize because unlike iterative local search methods, the moment computation step in Alg. 1 does not require much communication or synchronization.

There might be concerns about the claimed time efficiency, however, because significant performance improvements could only be observed when Spectral-sLDA is used together with Gibbs-sLDA, and the Gibbs sampling step might slow down the entire procedure. To see why this is not the case, we show in Table 2 that in order to obtain high-quality models and predictions, only a very small collection of documents are needed after model reconstruction of Alg. 1. In contrast, Gibbs-sLDA with random initialization requires more data to get reasonable performances.

To get a more intuitive idea of how fast our proposed method is, we combine Tables 1 and 2 to see that by doing Spectral-sLDA on $10^6$ documents and then post-processing the reconstructed models using Gibbs sampling on only $10^4$ documents, we obtain a pR$^2$ score of 0.17 in 5.8 minutes, while Gibbs-sLDA takes over an hour to process a million documents with a pR$^2$ score of only 0.14. Similarly, the hybrid method takes only 10 minutes to get a per-word likelihood comparable to the Gibbs sampling algorithm that requires more than an hour running time.

## 6 Conclusion

We propose a novel spectral decomposition based method to reconstruct supervised LDA models from labeled documents. Although our work has mainly focused on tensor decomposition based algorithms, it is an interesting problem whether NMF based methods could also be applied to obtain better sample complexity bound and superior performance in practice for supervised topic models.

## Acknowledgement

The work was done when Y.W. was at Tsinghua. The work is supported by the National Basic Research Program of China (No. 2013CB329403), National NSF of China (Nos. 61322308, 61332007), and Tsinghua University Initiative Scientific Research Program (No. 20121088071).

## Footnotes

[1] $c_i$ is a scalar coefficient that depends on $\alpha_0$ and $\alpha_i$. See Appendix A.2 for details.

[2]Due to the randomness in the data generating process, the true model has a non-zero prediction error.

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
