[Supplementary Material]

# Appendix A. Proof to Theorem 1

In this section, we prove the sample complexity bound given in Theorem 1. The proof consists of three main parts. In Appendix A.1, we prove perturbation lemmas that bound the estimation error of the whitened tensors $M_2(W, W)$, $M_y(W, W)$ and $M_3(W, W, W)$ in terms of the estimation error of the tensors themselves. In Appendix A.2, we cite results on the accuracy of SVD and robust tensor power method when performed on estimated tensors, and prove the effectiveness of the power update method used in recovering the linear regression model $\boldsymbol{\eta}$. Finally, we give tail bounds for the estimation error of $M_2$, $M_y$ and $M_3$ in Appendix A.3 and complete the proof in Appendix A.4. We also make some remarks on the indirect quantities (e.g. $\sigma_k(\widetilde{O})$) used in Theorem 1 and simplified bounds for some special cases in Appendix A.4.

All norms in the following analysis, if not explicitly specified, are 2 norms in the vector and matrix cases and the operator norm in the high-order tensor case.

## Appendix A.1. Perturbation lemmas

We first define the canonical topic distribution vectors $\widetilde{\boldsymbol{\mu}}$ and estimation error of observable tensors, which simplify the notations that arise in subsequent analysis.

**Definition 1** (canonical topic distribution). *Define the canonical version of topic distribution vector $\boldsymbol{\mu}_i$, $\widetilde{\boldsymbol{\mu}}_i$, as follows:*

$$\widetilde{\boldsymbol{\mu}}_i \triangleq \sqrt{\frac{\alpha_i}{\alpha_0(\alpha_0 + 1)}} \boldsymbol{\mu}_i. \tag{1}$$

*We also define $O, \widetilde{O} \in \mathbb{R}^{n \times k}$ by $O = [\boldsymbol{\mu}_1, \cdots, \boldsymbol{\mu}_k]$ and $\widetilde{O} = [\widetilde{\boldsymbol{\mu}_1}, \cdots, \widetilde{\boldsymbol{\mu}_k}]$.*

**Definition 2** (estimation error). *Assume*

$$\|M_2 - \widehat{M_2}\| \leq E_P, \tag{2}$$
$$\|M_y - \widehat{M_y}\| \leq E_y, \tag{3}$$
$$\|M_3 - \widehat{M_3}\| \leq E_T. \tag{4}$$

*for some real values $E_P, E_y$ and $E_T$, which we will set later.*

The following lemma analyzes the whitening matrix $W$ of $M_2$. Many conclusions are directly from [AFH$^+$12].

**Lemma 1** (Lemma C.1, [AFH$^+$12]). *Let $W, \widehat{W} \in \mathbb{R}^{n \times k}$ be the whitening matrices such that $M_2(W, W) = \widehat{M_2}(\widehat{W}, \widehat{W}) = I_k$. Let $A = W^\top \widetilde{O}$ and $\widehat{A} = \widehat{W}^\top \widetilde{O}$. Suppose $E_P \leq \sigma_k(M_2)/2$. We have*

$$\|W\| = \frac{1}{\sigma_k(\widetilde{O})}, \tag{5}$$

$$\|\widehat{W}\| \leq \frac{2}{\sigma_k(\widetilde{O})}, \tag{6}$$

$$\|W - \widehat{W}\| \leq \frac{4E_P}{\sigma_k(\widetilde{O})^3}, \tag{7}$$

$$\|W^+\| \leq 3\sigma_1(\widetilde{O}), \tag{8}$$
$$\|\widehat{W}^+\| \leq 2\sigma_1(\widetilde{O}), \tag{9}$$

$$\|W^+ - \widehat{W}^+\| \leq \frac{6\sigma_1(\widetilde{O})}{\sigma_k(\widetilde{O})^2} E_P, \tag{10}$$

$$\|A\| = 1, \tag{11}$$
$$\|\widehat{A}\| \leq 2, \tag{12}$$

$$\|A - \widehat{A}\| \;\leq\; \frac{4E_P}{\sigma_k(\widetilde{O})^2}, \tag{13}$$

$$\|AA^\top - \widehat{A}\widehat{A}^\top\| \;\leq\; \frac{12E_P}{\sigma_k(\widetilde{O})^2}. \tag{14}$$

*Proof.* Proof to Eq. (7): Let $\widehat{W}^\top \widehat{M_2}\widehat{W} = I$ and $\widehat{W}^\top M_2 \widehat{W} = BDB^\top$, where $B$ is orthogonal and $D$ is a positive definite diagonal matrix. We then see that $W = \widehat{W}BD^{-1/2}B^\top$ satisfies the condition $WM_2W^\top = I$. Subsequently, $\widehat{W} = WBD^{1/2}B^\top$. We then can bound $\|W - \widehat{W}\|$ as follows

$$\|W - \widehat{W}\| \leq \|W\| \cdot \|I - D^{1/2}\| \leq \|W\| \cdot \|I - D\| \leq \frac{4E_P}{\sigma_k(\widetilde{O})^3},$$

where the inequality $\|I - D\| \leq \frac{4E_P}{\sigma_k(\widetilde{O})^2}$ was proved in [AFH+12].

Proof to Eq. (14): $\|AA^\top - \widehat{A}\widehat{A}^\top\| \leq \|AA^\top - A\widehat{A}^\top\| + \|A\widehat{A}^\top - \widehat{A}\widehat{A}^\top\| \leq \|A - \widehat{A}\| \cdot (\|A\| + \|\widehat{A}\|) \leq \frac{12E_P}{\sigma_k(\widetilde{O})^2}$. All the other inequalities come from Lemma C.1, [AFH+12]. $\square$

We are now able to provide perturbation bounds for estimation error of whitened moments.

**Definition 3** (estimation error of whitened moments). *Define*

$$\varepsilon_{p,w} \;\triangleq\; \|M_2(W,W) - \widehat{M_2}(\widehat{W},\widehat{W})\|, \tag{15}$$

$$\varepsilon_{y,w} \;\triangleq\; \|M_y(W,W) - \widehat{M_y}(\widehat{W},\widehat{W})\|, \tag{16}$$

$$\varepsilon_{t,w} \;\triangleq\; \|M_3(W,W,W) - \widehat{M_3}(\widehat{W},\widehat{W},\widehat{W})\|. \tag{17}$$

**Lemma 2** (Perturbation lemma of whitened moments). *Suppose $E_P \leq \sigma_k(M_2)/2$. We have*

$$\varepsilon_{p,w} \;\leq\; \frac{16E_P}{\sigma_k(\widetilde{O})^2}, \tag{18}$$

$$\varepsilon_{y,w} \;\leq\; \frac{24\|\boldsymbol{\eta}\|E_P}{(\alpha_0+2)\sigma_k(\widetilde{O})^2} + \frac{4E_y}{\sigma_k(\widetilde{O})^2}, \tag{19}$$

$$\varepsilon_{t,w} \;\leq\; \frac{54E_P}{(\alpha_0+1)(\alpha_0+2)\sigma_k(\widetilde{O})^5} + \frac{8E_T}{\sigma_k(\widetilde{O})^3}. \tag{20}$$

*Proof.* Using the idea in the proof of Lemma C.2 in [AFH+12], we can split $\varepsilon_{p,w}$ as

$$\begin{aligned} \varepsilon_{p,w} &= \|M_2(W,W) - M_2(\widehat{W},\widehat{W}) + M_2(\widehat{W},\widehat{W}) - \widehat{M_2}(\widehat{W},\widehat{W})\| \\ &\leq \|M_2(W,W) - M_2(\widehat{W},\widehat{W})\| + \|M_2(\widehat{W},\widehat{W}) - \widehat{M_2}(\widehat{W},\widehat{W})\|. \end{aligned}$$

We can the bound the two terms seperately, as follows.

For the first term, we have

$$\begin{aligned} \|M_2(W,W) - M_2(\widehat{W},\widehat{W})\| &= \|W^\top M_2 W - \widehat{W}^\top \widehat{M_2}\widehat{W}\| \\ &= \|AA^\top - \widehat{A}\widehat{A}^\top\| \\ &\leq \frac{12E_P}{\sigma_k(\widetilde{O})^2}. \end{aligned}$$

where the last inequality comes from Eq. (14)

For the second term, we have

$$\|M_2(\widehat{W},\widehat{W}) - \widehat{M_2}(\widehat{W},\widehat{W})\| \leq \|\widehat{W}\|^2 \cdot \|M_2 - \widehat{M_2}\| \leq \frac{4E_P}{\sigma_k(\widetilde{O})^2},$$

where the last inequality comes from Eq. (6).

Similarly, $\varepsilon_{y,w}$ can be splitted as $\|M_y(W,W) - M_y(\widehat{W}, \widehat{W})\|$ and $\|M_y(\widehat{W}, \widehat{W}) - \widehat{M}_y(\widehat{W}, \widehat{W})\|$, which can be bounded separately.

For the first term, we have

$$
\begin{aligned}
\|M_y(W,W) - M_y(\widehat{W}, \widehat{W})\| &= \|W^\top M_y W - \widehat{W}^\top M_y \widehat{W}\| \\
&= \frac{2}{\alpha_0 + 2} \|A \mathrm{diag}(\boldsymbol{\eta}) A^\top - \widehat{A} \mathrm{diag}(\boldsymbol{\eta}) \widehat{A}^\top\| \\
&\leq \frac{2\|\boldsymbol{\eta}\|}{\alpha_0 + 2} \cdot \|AA^\top - \widehat{A}\widehat{A}^\top\| \\
&\leq \frac{24\|\boldsymbol{\eta}\|}{(\alpha_0 + 2)\sigma_k(\widetilde{O})^2} \cdot E_P.
\end{aligned}
$$

For the second term, we have

$$
\|M_y(\widehat{W}, \widehat{W}) - \widehat{M}_y(\widehat{W}, \widehat{W})\| \leq \|\widehat{W}\|^2 \cdot \|M_y - \widehat{M}_y\| \leq \frac{4E_y}{\sigma_k(\widetilde{O})^2}.
$$

Finally, we bound $\varepsilon_{t,w}$ as below, following the work [CL13].

$$
\begin{aligned}
\varepsilon_{t,w} &= \|M_3(W,W,W) - \widehat{M}_3(\widehat{W}, \widehat{W}, \widehat{W})\| \\
&\leq \|M_3\| \cdot \|W - \widehat{W}\| \cdot (\|W\|^2 + \|W\| \cdot \|\widehat{W}\| + \|\widehat{W}\|^2) + \|\widehat{W}\|^3 \cdot \|M_3 - \widehat{M}_3\| \\
&\leq \frac{54 E_P}{(\alpha_0 + 1)(\alpha_0 + 2)\sigma_k(\widetilde{O})^5} + \frac{8E_T}{\sigma_k(\widetilde{O})^3},
\end{aligned}
$$

where we have used the fact that $\|M_3\| \leq \sum_{i=1}^k \frac{2\alpha_i}{\alpha_0(\alpha_0+1)(\alpha_0+2)} = \frac{2}{(\alpha_0+1)(\alpha_0+2)}$.  $\square$

## Appendix A.2. SVD accuracy

The key idea for spectral recovery of LDA topic modeling is the *simultaneous diagonalization* trick, which asserts that we can recover LDA model parameters by performing orthogonal tensor decomposition on a pair of simultaneously whitened moments, for example, $(M_2, M_3)$ and $(M_2, M_y)$. The following proposition details this insight, as we derive orthogonal tensor decompositions for the whitened tensor product $M_y(W,W)$ and $M_3(W,W,W)$.

**Proposition 1.** *Define $\boldsymbol{v}_i \triangleq W^\top \widetilde{\boldsymbol{\mu}}_i = \sqrt{\frac{\alpha_i}{\alpha_0(\alpha_0+1)}} W^\top \boldsymbol{\mu}_i$. Then*

1. *$\{\boldsymbol{v}_i\}_{i=1}^k$ is an orthonormal basis.*

2. *$M_y$ has a pair of singular value and singular vector $(\sigma_i^y, \boldsymbol{v}_i)$ with $\sigma_i^y = \frac{2}{\alpha_0+2}\eta_j$ for some $j \in [k]$.*

3. *$M_3$ has a pair of robust eigenvalue and eigenvector [AGH$^+$12] $(\lambda_i, \boldsymbol{v}_i)$ with $\lambda_i = \frac{2}{\alpha_0+2}\sqrt{\frac{\alpha_0(\alpha_0+1)}{\alpha_{j'}}}$ for some $j' \in [k]$.*

*Proof.* The orthonormality of $\{\boldsymbol{v}_i\}_{i=1}^k$ follows from the fact that $W^\top M_2 W = \sum_{i=1}^k \boldsymbol{v}_i \boldsymbol{v}_i^\top = I_k$. Subsequently, we have

$$
M_y(W,W) = \frac{2}{\alpha_0+2} \sum_{i=1}^k \eta_i \boldsymbol{v}_i \boldsymbol{v}_i^\top,
$$

$$
M_3(W,W,W) = \frac{2}{\alpha_0+2} \sum_{i=1}^k \sqrt{\frac{\alpha_0(\alpha_0+1)}{\alpha_i}} \boldsymbol{v}_i \otimes \boldsymbol{v}_i \otimes \boldsymbol{v}_i.
$$

$\square$

The following lemmas (Lemma 3 and Lemma 4) give upper bounds on the estimation error of $\boldsymbol{\eta}$ and $\boldsymbol{\mu}$ in terms of $|\widehat{\lambda}_i - \lambda_i|$, $|\widehat{\boldsymbol{v}}_i - \boldsymbol{v}_i|$ and the estimation errors of whitened moments defined in Definition 3.

**Lemma 3** ($\eta_i$ estimation error bound)**.** *Define* $\widehat{\eta}_i \triangleq \frac{\alpha_0+2}{2}\widehat{\boldsymbol{v}}_i^\top \widehat{M}_y(\widehat{W}, \widehat{W})\widehat{\boldsymbol{v}}_i$, *where* $\widehat{\boldsymbol{v}}_i$ *is some estimation of* $\boldsymbol{v}_i$. *We then have*

$$|\eta_i - \widehat{\eta}_i| \le 2\|\boldsymbol{\eta}\|\|\widehat{\boldsymbol{v}}_i - \boldsymbol{v}_i\| + \frac{\alpha_0+2}{2}(1 + 2\|\widehat{\boldsymbol{v}}_i - \boldsymbol{v}_i\|) \cdot \varepsilon_{y,w}. \tag{21}$$

*Proof.* First, note that $\boldsymbol{v}_i^\top M_y(W,W)\boldsymbol{v}_i = \frac{2}{\alpha_0+2}\eta_i$ because $\{\boldsymbol{v}_i\}_{i=1}^k$ are orthonormal. Subsequently, we have

$$\begin{aligned}\frac{2}{\alpha_0+2}|\eta_i - \widehat{\eta}_i| &= \left|\widehat{\boldsymbol{v}}_i^\top \widehat{M}_y(\widehat{W},\widehat{W})\widehat{\boldsymbol{v}}_i - \boldsymbol{v}_i^\top M_y(W,W)\boldsymbol{v}_i\right| \\ &\le \left|(\widehat{\boldsymbol{v}}_i - \boldsymbol{v}_i)^\top \widehat{M}_y(\widehat{W},\widehat{W})\widehat{\boldsymbol{v}}_i\right| + \left|\boldsymbol{v}_i^\top\left(\widehat{M}_y(\widehat{W},\widehat{W})\widehat{\boldsymbol{v}}_i - M_y(W,W)\boldsymbol{v}_i\right)\right| \\ &\le \|\widehat{\boldsymbol{v}}_i - \boldsymbol{v}_i\|\|\widehat{M}_y(\widehat{W},\widehat{W})\|\|\widehat{\boldsymbol{v}}_i\| + \|\boldsymbol{v}_i\|\|\widehat{M}_y(\widehat{W},\widehat{W})\widehat{\boldsymbol{v}}_i - M_y(W,W)\boldsymbol{v}_i\|.\end{aligned}$$

Note that both $\boldsymbol{v}_i$ and $\widehat{\boldsymbol{v}}_i$ are unit vectors. Therefore,

$$\begin{aligned}\frac{2}{\alpha_0+2}|\eta_i - \widehat{\eta}_i| &\le \|\widehat{M}_y(\widehat{W},\widehat{W})\|\|\widehat{\boldsymbol{v}}_i - \boldsymbol{v}_i\| + \|\widehat{M}_y(\widehat{W},\widehat{W})\widehat{\boldsymbol{v}}_i - M_y(W,W)\boldsymbol{v}_i\| \\ &\le \|\widehat{M}_y(\widehat{W},\widehat{W})\|\|\widehat{\boldsymbol{v}}_i - \boldsymbol{v}_i\| + \|\widehat{M}_y(\widehat{W},\widehat{W})\|\|\widehat{\boldsymbol{v}}_i - \boldsymbol{v}_i\| + \|\widehat{M}_y(\widehat{W},\widehat{W}) - M_y(W,W)\|\|\boldsymbol{v}_i\| \\ &\le 2\|\widehat{\boldsymbol{v}}_i - \boldsymbol{v}_i\|\left(\frac{2}{\alpha_0+2}\|\boldsymbol{\eta}\| + \varepsilon_{y,w}\right) + \varepsilon_{y,w}.\end{aligned}$$

The last inequality is due to the fact that $\|M_y(W,W)\| = \frac{2}{\alpha_0+2}\|\boldsymbol{\eta}\|$.  $\square$

**Lemma 4** ($\boldsymbol{\mu}_i$ estimation error bound)**.** *Define* $\widehat{\boldsymbol{\mu}}_i \triangleq \frac{\alpha_0+2}{2}\widehat{\lambda}_i(\widehat{W}^+)^\top\widehat{\boldsymbol{v}}_i$, *where* $\widehat{\lambda}_i, \widehat{\boldsymbol{v}}_i$ *are some estimates of singular value pairs* $(\lambda_i, \boldsymbol{v}_i)$ *of* $M_3(W,W,W)$. *We then have*

$$\|\widehat{\boldsymbol{\mu}}_i - \boldsymbol{\mu}_i\| \le \frac{3(\alpha_0+2)}{2}\sigma_1(\widetilde{O})|\widehat{\lambda}_i - \lambda_i| + 3\alpha_{\max}\sigma_1(\widetilde{O})\|\widehat{\boldsymbol{v}}_i - \boldsymbol{v}_i\| + \frac{6\alpha_{\max}\sigma_1(\widetilde{O})E_P}{\sigma_k(\widetilde{O})^2}. \tag{22}$$

*Proof.* First note that $\boldsymbol{\mu}_i = \frac{\alpha_0+2}{2}\lambda_i(W^+)^\top\boldsymbol{v}_i$. Subsequently,

$$\begin{aligned}\frac{2}{\alpha_0+2}\|\boldsymbol{\mu}_i - \widehat{\boldsymbol{\mu}}_i\| &= \|\widehat{\lambda}_i(\widehat{W}^+)^\top\widehat{\boldsymbol{v}}_i - \lambda_i(W^+)^\top\boldsymbol{v}_i\| \\ &\le \|\widehat{\lambda}_i\widehat{W}^+ - \lambda_iW^+\|\|\widehat{\boldsymbol{v}}_i\| + \|\lambda_iW^+\|\|\widehat{\boldsymbol{v}}_i - \boldsymbol{v}_i\| \\ &\le |\widehat{\lambda}_i - \lambda_i|\|\widehat{W}^+\| + |\lambda_i|\|\widehat{W}^+ - W^+\| + |\lambda_i|\|W^+\|\|\widehat{\boldsymbol{v}}_i - \boldsymbol{v}_i\| \\ &\le 3\sigma_1(\widetilde{O})|\widehat{\lambda}_i - \lambda_i| + \frac{2\alpha_{\max}}{\alpha_0+2}\cdot\frac{6\sigma_1(\widetilde{O})E_P}{\sigma_k(\widetilde{O})^2} + \frac{2\alpha_{\max}}{\alpha_0+2}\cdot 3\sigma_1(\widetilde{O})\cdot\|\widehat{\boldsymbol{v}}_i - \boldsymbol{v}_i\|.\end{aligned}$$

$\square$

To bound the error of orthogonal tensor decomposition performed on the estimated tensors $\widehat{M}_3(\widehat{W},\widehat{W},\widehat{W})$, we cite Theorem 5.1 [AGH$^+$12], a sample complexity analysis on the robust tensor power method we used for recovering $\widehat{\lambda}_i$ and $\widehat{\boldsymbol{v}}_i$.

**Lemma 5** (Theorem 5.1, [AGH$^+$12]). *Let $\lambda_{\max} = \frac{2}{\alpha_0+2}\sqrt{\frac{\alpha_0(\alpha_0+1)}{\alpha_{\min}}}$, $\lambda_{\min} = \frac{2}{\alpha_0+2}\sqrt{\frac{\alpha_0(\alpha_0+1)}{\alpha_{\max}}}$, where $\alpha_{\min} = \min \alpha_i$ and $\alpha_{\max} = \max \alpha_i$. Then there exist universal constants $C_1, C_2 > 0$ such that the following holds: Fix $\delta' \in (0, 1)$. Suppose $\varepsilon_{t,w} \le \varepsilon$ and*

$$\varepsilon_{t,w} \quad \le \quad C_1 \cdot \frac{\lambda_{\min}}{k}, \tag{23}$$

*Suppose $\{(\widehat{\lambda}_i, \widehat{\boldsymbol{v}}_i)\}_{i=1}^k$ are eigenvalue and eigenvector pairs returned by running Algorithm 1 in [AGH$^+$12] with input $\widehat{M}_3(\widehat{W}, \widehat{W}, \widehat{W})$ for $L = poly(k)\log(1/\delta')$ and $N \ge C_2 \cdot (\log(k) + \log\log(\frac{\lambda_{\max}}{\varepsilon}))$ iterations. With probability greater than $1 - \delta'$, there exists a permutation $\pi' : [k] \to [k]$ such that for all $i$,*

$$\|\widehat{\boldsymbol{v}}_i - \boldsymbol{v}_{\pi'(i)}\| \le 8\varepsilon/\lambda_{\min}, \quad |\widehat{\lambda}_i - \lambda_{\pi'(i)}| \le 5\varepsilon.$$

## Appendix A.3. Tail Inequalities

**Lemma 6** (Lemma 5, [CL13]). *Let $\boldsymbol{x}_1, \cdots, \boldsymbol{x}_N \in \mathbb{R}^d$ be i.i.d. samples from some distribution with bounded support (i.e., $\|\boldsymbol{x}\|_2 \le B$ with probability 1 for some constant B). Then with probability at least $1 - \delta$,*

$$\left\| \frac{1}{N} \sum_{i=1}^N \boldsymbol{x}_i - \mathbb{E}[\boldsymbol{x}] \right\|_2 \le \frac{2B}{\sqrt{N}}\left( 1 + \sqrt{\frac{\log(1/\delta)}{2}} \right).$$

**Corrolary 1.** *Let $\boldsymbol{x}_1, \cdots, \boldsymbol{x}_N \in \mathbb{R}^d$ be i.i.d. samples from some distributions with $\Pr[\|\boldsymbol{x}\|_2 \le B] \ge 1 - \delta'$. Then with probability at least $1 - N\delta' - \delta$,*

$$\left\| \frac{1}{N} \sum_{i=1}^N \boldsymbol{x}_i - \mathbb{E}[\boldsymbol{x}] \right\|_2 \le \frac{2B}{\sqrt{N}}\left( 1 + \sqrt{\frac{\log(1/\delta)}{2}} \right).$$

*Proof.* Use union bound. $\qquad\qquad\qquad\qquad\qquad\qquad\qquad\qquad\qquad\qquad\qquad\qquad\qquad\square$

**Lemma 7** (concentraion of moment norms). *Suppose we obtain N i.i.d. samples (i.e., documents with at least three words each and their regression variables in sLDA models). Define $R(\delta) \triangleq \|\boldsymbol{\eta}\| + \Phi^{-1}(\delta)$, where $\Phi^{-1}(\cdot)$ is the inverse function of the CDF of a standard Gaussian distribution. Let $\mathbb{E}[\cdot]$ denote the mean of the true underlying distribution and $\widehat{\mathbb{E}}[\cdot]$ denote the empirical mean. Then*

*1.* $\Pr\left[ \|\mathbb{E}[\boldsymbol{x}_1] - \widehat{\mathbb{E}}[\boldsymbol{x}_1]\|_F < \frac{2+\sqrt{2\log(1/\delta)}}{\sqrt{N}} \right] \ge 1 - \delta.$

*2.* $\Pr\left[ \|\mathbb{E}[\boldsymbol{x}_1 \otimes \boldsymbol{x}_2] - \widehat{\mathbb{E}}[\boldsymbol{x}_1 \otimes \boldsymbol{x}_2]\|_F < \frac{2+\sqrt{2\log(1/\delta)}}{\sqrt{N}} \right] \ge 1 - \delta.$

*3.* $\Pr\left[ \|\mathbb{E}[\boldsymbol{x}_1 \otimes \boldsymbol{x}_2 \otimes \boldsymbol{x}_3] - \widehat{\mathbb{E}}[\boldsymbol{x}_1 \otimes \boldsymbol{x}_2 \otimes \boldsymbol{x}_3]\|_F < \frac{2+\sqrt{2\log(1/\delta)}}{\sqrt{N}} \right] \ge 1 - \delta.$

*4.* $\Pr\left[ \|\mathbb{E}[y] - \widehat{\mathbb{E}}[y]\| < R(\delta/4\sigma N) \cdot \frac{2+\sqrt{2\log(2/\delta)}}{\sqrt{N}} \right] \ge 1 - \delta.$

*5.* $\Pr\left[ \|\mathbb{E}[y\boldsymbol{x}_1] - \widehat{\mathbb{E}}[y\boldsymbol{x}_1]\|_F < R(\delta/4\sigma N) \cdot \frac{2+\sqrt{2\log(2/\delta)}}{\sqrt{N}} \right] \ge 1 - \delta.$

*6.* $\Pr\left[ \|\mathbb{E}[y\boldsymbol{x}_1 \otimes \boldsymbol{x}_2] - \widehat{\mathbb{E}}[y\boldsymbol{x}_1 \otimes \boldsymbol{x}_2]\|_F < R(\delta/4\sigma N) \cdot \frac{2+\sqrt{2\log(2/\delta)}}{\sqrt{N}} \right] \ge 1 - \delta.$

*Proof.* Use Lemma 6 and Corrolary 1 for concentration bounds involving the regression variable $y$. $\qquad\square$

**Corrolary 2.** *With probability $1 - \delta$ the following holds:*

1. $E_P = \|M_2 - \widehat{M_2}\| \le 3 \cdot \frac{2+\sqrt{2\log(6/\delta)}}{\sqrt{N}}.$

2. $E_y = \|M_y - \widehat{M_y}\| \le 10R(\delta/60\sigma N) \cdot \frac{2+\sqrt{2\log(15/\delta)}}{\sqrt{N}}.$

3. $E_T = \|M_3 - \widehat{M_3}\| \le 10 \cdot \frac{2+\sqrt{2\log(9/\delta)}}{\sqrt{N}}.$

*Proof.* Corrolary 2 can be proved by expanding the terms by definition and then using tail inequality in Lemma 7 and union bound. Also note that $\|\cdot\| \le \|\cdot\|_F$ for all matrices. $\qquad\square$

## Appendix A.4. Completing the proof

We are now ready to give a complete proof to Theorem 1.

**Theorem 1** (Sample complexity bound)**.** *Let $\sigma_1(\widetilde{O})$ and $\sigma_k(\widetilde{O})$ be the largest and the smallest singular values of the canonical topic distribution matrix $\widetilde{O}$. Define $\lambda_{\min} \triangleq \frac{2}{\alpha_0+2}\sqrt{\frac{\alpha_0(\alpha_0+1)}{\alpha_{\max}}}$ and $\lambda_{\max} \triangleq \frac{2}{\alpha_0+2}\sqrt{\frac{\alpha_0(\alpha_0+1)}{\alpha_{\min}}}$ where $\alpha_{\max}$ and $\alpha_{\min}$ are the largest and the smallest entries in $\boldsymbol{\alpha}$. Suppose $\widehat{\boldsymbol{\mu}}$, $\widehat{\boldsymbol{\alpha}}$ and $\widehat{\boldsymbol{\eta}}$ are the outputs of Algorithm 1, and $L$ is at least a linear function in $k$. Fix $\delta \in (0,1)$. For any small error-tolerance parameter $\varepsilon > 0$, if Algorithm 1 is run with parameter $T = \Omega(\log(k) + \log\log(\lambda_{\max}/\varepsilon))$ on $N$ i.i.d. sampled documents with $N \ge \max(n_1, n_2, n_3)$, where*

$$n_1 = C_1 \cdot \left(1 + \sqrt{\log(6/\delta)}\right)^2 \cdot \frac{\alpha_0^2(\alpha_0+1)^2}{\alpha_{\min}}, \tag{24}$$

$$n_2 = C_2 \cdot \frac{(1 + \sqrt{\log(15/\delta)})^2}{\varepsilon^2 \sigma_k(\widetilde{O})^4} \cdot \max\left((\|\boldsymbol{\eta}\| + \Phi^{-1}(\delta/60\sigma))^2, \alpha_{\max}^2\sigma_1(\widetilde{O})^2\right), \tag{25}$$

$$n_3 = C_3 \cdot \frac{(1 + \sqrt{\log(9/\delta)})^2}{\sigma_k(\widetilde{O})^{10}} \cdot \max\left(\frac{1}{\varepsilon^2}, \frac{k^2}{\lambda_{\min}^2}\right), \tag{26}$$

*and $C_1, C_2$ and $C_3$ are universal constants, then with probability at least $1-\delta$, there exists a permutation $\pi : [k] \to [k]$ such that for every topic $i$, the following holds:*

1. $|\alpha_i - \widehat{\alpha}_{\pi(i)}| \le \frac{4\alpha_0(\alpha_0+1)(\lambda_{\max}+5\varepsilon)}{(\alpha_0+2)^2\lambda_{\min}^2(\lambda_{\min}-5\varepsilon)^2} \cdot 5\varepsilon,$ *if $\lambda_{\min} > 5\varepsilon$;*

2. $\|\boldsymbol{\mu}_i - \widehat{\boldsymbol{\mu}}_{\pi(i)}\| \le \left(3\sigma_1(\widetilde{O})\left(\frac{8\alpha_{\max}}{\lambda_{\min}} + \frac{5(\alpha_0+2)}{2}\right) + 1\right)\varepsilon;$

3. $|\eta_i - \widehat{\eta}_{\pi(i)}| \le \left(\frac{\|\boldsymbol{\eta}\|}{\lambda_{\min}} + (\alpha_0+2)\right)\varepsilon.$

*Proof.* First, the assumption $E_P \le \sigma_k(M_2)$ is required for error bounds on $\varepsilon_{p,w}, \varepsilon_{y,w}$ and $\varepsilon_{t,w}$. Noting Corrolary 2 and the fact that $\sigma_k(M_2) = \frac{\alpha_{\min}}{\alpha_0(\alpha_0+1)}$, we have

$$N = \Omega\left(\frac{\alpha_0^2(\alpha_0+1)^2(1+\sqrt{\log(6/\delta)})^2}{\alpha_{\min}^2}\right).$$

Note that this lower bound does not depend on $k$, $\varepsilon$ and $\sigma_k(\widetilde{O})$.

For Lemma 5 to hold, we need the assumptions that $\varepsilon_{t,w} \le \min(\varepsilon, O(\frac{\lambda_{\min}}{k}))$. These imply Eq. (26), as we expand $\varepsilon_{t,w}$ according to Definition 3 and note the fact that the first term $\frac{54E_P}{(\alpha_0+1)(\alpha_0+2)\sigma_k(\widetilde{O})^5}$

dominates the second one. The $\alpha_0$ is missing in Eq. (26) because $\alpha_0 + 1 \geq 1$, $\alpha_0 + 2 \geq 2$ and we discard them both. The $|\alpha_i - \widehat{\alpha}_{\pi(i)}|$ bound follows immediately by Lemma 5 and the recovery rule $\widehat{\alpha}_i = \frac{\alpha_0 + 2}{2}\widehat{\lambda}_i$.

To bound the estimation error for the linear classifier $\boldsymbol{\eta}$, we need to further bound $\varepsilon_{y,w}$. We assume $\varepsilon_{y,w} \leq \varepsilon$. By expanding $\varepsilon_{y,w}$ according to Definition 3 in a similar manner we obtain the $(\|\boldsymbol{\eta}\| + \Phi^{-1}(\delta/60\sigma))^2$ term in Eq. (25). The bound on $|\eta_i - \widehat{\eta}_{\pi(i)}|$ follows immediately by Lemma 3.

Finally, we bound $\|\boldsymbol{\mu}_i - \widehat{\boldsymbol{\mu}}_{\pi(i)}\|$ using Lemma 4. We need to assume that $\frac{6\alpha_{\max}\sigma_1(\widetilde{O})E_P}{\sigma_k(\widetilde{O})^2} \leq \varepsilon$, which gives the $\alpha_{\max}^2\sigma_1(\widetilde{O})^2$ term in Eq. (25). The $\|\boldsymbol{\mu}_i - \widehat{\boldsymbol{\mu}}_{\pi(i)}\|$ bound then follows by Lemma 4 and Lemma 5. $\qquad \square$

We make some remarks for the main theorem. In Remark 1, we establish links between indirect quantities appeared in Theorem 1 (e.g., $\sigma_k(\widetilde{O})$) and the functions of original model parameters (e.g., $\sigma_k(O)$). These connections are straightforward following their definitions.

**Remark 1.** *The indirect quantities $\sigma_1(\widetilde{O})$ and $\sigma_k(\widetilde{O})$ can be related to $\sigma_1(O)$, $\sigma_k(O)$ and $\boldsymbol{\alpha}$ in the following way:*

$$\sqrt{\frac{\alpha_{\min}}{\alpha_0(\alpha_0 + 1)}}\sigma_k(O) \leq \sigma_k(\widetilde{O}) \leq \sqrt{\frac{\alpha_{\max}}{\alpha_0(\alpha_0 + 1)}}\sigma_k(O);$$

$$\sigma_1(\widetilde{O}) \leq \sqrt{\frac{\alpha_{\max}}{\alpha_0(\alpha_0 + 1)}}\sigma_1(O) \leq \frac{1}{\sqrt{\alpha_0 + 1}}.$$

We now take a close look at the sample complexity bound in Theorem 1. It is evident that $n_2$ can be neglected when the number of topics $k$ gets large, because in practice the norm of the linear regression model $\boldsymbol{\eta}$ is usually assumed to be small in order to avoid overfitting. Moreover, as mentioned before, the prior parameter $\boldsymbol{\alpha}$ is often assumed to be homogeneous with $\alpha_i = 1/k$ [SG07]. With these observations, the sample complexity bound in Theorem 1 can be greatly simplified.

**Remark 2.** *Assume $\|\boldsymbol{\eta}\|$ and $\sigma$ are small and $\boldsymbol{\alpha} = (1/k, \cdots, 1/k)$. As the number of topics $k$ gets large, the sample complexity bound in Theorem 1 can be simplified as*

$$N = \Omega\left(\frac{\log(1/\delta)}{\sigma_k(\widetilde{O})^{10}} \cdot \max(\varepsilon^{-2}, k^3)\right). \tag{27}$$

The sample complexity bound in Remark 2 may look formidable as it depends on $\sigma_k(\widetilde{O})^{10}$. However, such dependency is somewhat necessary because we are using third-order tensors to recover the underlying model parameters. Furthermore, the dependence on $\sigma_k(\widetilde{O})^{10}$ is introduced by the robust tensor power method to recover LDA parameters, and the reconstruction accuracy of $\boldsymbol{\eta}$ only depends on $\sigma_k(\widetilde{O})^4$ and $(\|\boldsymbol{\eta}\| + \Phi^{-1}(\delta/60\sigma))^2$. As a consequence, if we can combine our power update method for $\boldsymbol{\eta}$ with LDA inference algorithms that have milder dependence on the singular value $\sigma_k(\widetilde{O})$, we might be able to get an algorithm with a better sample complexity. Such an extension is discussed in Appendix C.1.

# Appendix B. Moments of Observable Variables

## Appendix B.1. Proof to Proposition 1

The equations on $M_2$ and $M_3$ have already been proved in [AFH$^+$12] and [AGH$^+$12]. Here we only give the proof to the equation on $M_y$. In fact, all the three equations can be proved in a similar manner.

In sLDA the topic mixing vector $\boldsymbol{h}$ follows a Dirichlet prior distribution with parameter $\boldsymbol{\alpha}$. Therefore, we have

$$\mathbb{E}[h_i] = \frac{\alpha_i}{\alpha_0}, \mathbb{E}[h_i h_j] = \begin{cases} \frac{\alpha_i^2}{\alpha_0(\alpha_0 + 1)}, & \text{if } i = j, \\ \frac{\alpha_i \alpha_j}{\alpha_0^2}, & \text{if } i \neq j \end{cases}, \mathbb{E}[h_i h_j h_k] = \begin{cases} \frac{\alpha_i^3}{\alpha_0(\alpha_0+1)(\alpha_0+2)}, & \text{if } i = j = k, \\ \frac{\alpha_i^2 \alpha_k}{\alpha_0^2(\alpha_0+1)}, & \text{if } i = j \neq k, \\ \frac{\alpha_i \alpha_j \alpha_k}{\alpha_0^3}, & \text{if } i \neq j, j \neq k, i \neq k \end{cases} \tag{28}$$

Next, note that

$$\mathbb{E}[y|\boldsymbol{h}] = \boldsymbol{\eta}^\top \boldsymbol{h}, \ \mathbb{E}[\boldsymbol{x}_1|\boldsymbol{h}] = \sum_{i=1}^{k} h_i \boldsymbol{\mu}_i, \ \mathbb{E}[\boldsymbol{x}_1 \otimes \boldsymbol{x}_2|\boldsymbol{h}] = \sum_{i,j=1}^{k} h_i h_j \boldsymbol{\mu}_i \otimes \boldsymbol{\mu}_j, \tag{29}$$

$$\mathbb{E}[y\boldsymbol{x}_1 \otimes \boldsymbol{x}_2|\boldsymbol{h}] = \sum_{i,j,k=1}^{k} h_i h_j h_k \cdot \eta_k \boldsymbol{\mu}_j \otimes \boldsymbol{\mu}_k. \tag{30}$$

Proposition 1 can then be proved easily by taking expectation over the topic mixing vector $\boldsymbol{h}$.

## Appendix B.2. Details of the speeding-up trick

In this section we provide details of the trick mentioned in the main paper to speed up empirical moments computations. First, note that the computation of $\widehat{M}_1$, $\widehat{M}_2$ and $\widehat{M}_y$ only requires $O(NM^2)$ time and $O(V^2)$ space. They do not need to be accelerated in most practical applications. This time and space complexity also applies to all terms in $\widehat{M}_3$ except the $\widehat{\mathbb{E}}[\boldsymbol{x}_1 \otimes \boldsymbol{x}_2 \otimes \boldsymbol{x}_3]$ term, which requires $O(NM^3)$ time and $O(V^3)$ space if using naive implementations. Therefore, this section is devoted to speed-up the computation of $\widehat{\mathbb{E}}[\boldsymbol{x}_1 \otimes \boldsymbol{x}_2 \otimes \boldsymbol{x}_3]$. More precisely, as mentioned in the main paper, what we want to compute is the whitened empirical moment $\widehat{\mathbb{E}}[\boldsymbol{x}_1 \otimes \boldsymbol{x}_2 \otimes \boldsymbol{x}_3](\widehat{W}, \widehat{W}, \widehat{W}) \in \mathbb{R}^{k \times k \times k}$.

Fix a document $D$ with $m$ words. Let $T \triangleq \widehat{\mathbb{E}}[\boldsymbol{x}_1 \otimes \boldsymbol{x}_2 \otimes \boldsymbol{x}_3|D]$ be the empirical tensor demanded. By definition, we have

$$T_{i,j,k} = \frac{1}{m(m-1)(m-2)} \begin{cases} n_i(n_j-1)(n_k-2), & i=j=k; \\ n_i(n_i-1)n_k, & i=j, j \neq k; \\ n_i n_j(n_j-1), & j=k, i \neq j; \\ n_i n_j(n_i-1), & i=k, i \neq j; \\ n_i n_j n_k, & \text{otherwise}; \end{cases} \tag{31}$$

where $n_i$ is the number of occurrences of the $i$-th word in document $D$. If $T_{i,j,k} = \frac{n_i n_j n_k}{m(m-1)(m-2)}$ for all indices $i, j$ and $k$, then we only need to compute

$$T(W, W, W) = \frac{1}{m(m-1)(m-2)} \cdot (W\boldsymbol{n})^{\otimes 3},$$

where $\boldsymbol{n} \triangleq (n_1, n_2, \cdots, n_V)$. This takes $O(Mk + k^3)$ computational time because $\boldsymbol{n}$ contains at most $M$ non-zero entries, and the total time complexity is reduced from $O(NM^3)$ to $O(N(Mk + k^3))$.

We now consider the remaining values, where at least two indices are identical. We first consider those values with two indices the same, for example, $i = j$. For these indices, we need to subtract an $n_i n_k$ term, as shown in Eq. (31). That is, we need to compute the whitened tensor $\Delta(W, W, W)$, where $\Delta \in \mathbb{R}^{V \times V \times V}$ and

$$\Delta_{i,j,k} = \frac{1}{m(m-1)(m-2)} \cdot \begin{cases} n_i n_k, & i=j; \\ 0, & \text{otherwise}. \end{cases} \tag{32}$$

Note that $\Delta$ can be written as $\frac{1}{m(m-1)(m-2)} \cdot A \otimes \boldsymbol{n}$, where $A = \text{diag}(n_1, n_2, \cdots, n_V)$ is a $V \times V$ matrix and $\boldsymbol{n} = (n_1, n_2, \cdots, n_V)$ is defined previously. As a result, $\Delta(W, W, W) = \frac{1}{m(m-1)(m-2)} \cdot (W^\top A W) \otimes \boldsymbol{n}$. So the computational complexity of $\Delta(W, W, W)$ depends on how we compute $W^\top A W$. Since $A$ is a diagonal matrix with at most $M$ non-zero entries, $W^\top A W$ can be computed in $O(Mk^2)$ operations. Therefore, the time complexity of computing $\Delta(W, W, W)$ is $O(Mk^2)$ per document.

Finally we handle those values with three indices the same, that is, $i = j = k$. As indicated by Eq. (31), we need to add a $\frac{2n_i}{m(m-1)(m-2)}$ term for compensation. This can be done efficiently by first computing $\widehat{\mathbb{E}}(\frac{2n_i}{m(m-1)(m-2)})$ for all the documents (requiring $O(NV)$ time), and then add them up, which takes $O(Vk^3)$ operations.

# Appendix C. Discussions

## Appendix C.1. Extension to other topic recovery algorithms

One important advantage of our proposed inference algorithm is its flexibility—the algorithm can be combined with many other LDA inference algorithms to infer supervised LDA model parameters. More specifically, given access to any algorithm that recovers the topic distribution matrix $O$ and the prior parameter $\boldsymbol{\alpha}$ from i.i.d. sampled documents, an inference algorithm for a supervised LDA model can be immediately obtained, as shown in Algorithm 1.

---

**Algorithm 1** sLDA parameter recovery based on an existing LDA inference algorithm $\mathcal{A}$. Input parameter: $\alpha_0$.

---

1: Compute empirical moments and obtain $\widehat{M_2}, \widehat{M_y}$. Set $\widehat{\boldsymbol{\eta}} = 0$.
2: Find $\widehat{W} \in \mathbb{R}^{n \times k}$ such that $\widehat{M_2}(\widehat{W}, \widehat{W}) = I_k$.
3: Run algorithm $\mathcal{A}$ with observed documents and parameter $\alpha_0$. Obtain the estimated topic distribution matrix $\widehat{O} = (\widehat{\boldsymbol{\mu}}_1, \cdots, \widehat{\boldsymbol{\mu}}_k)$ and prior parameter $\widehat{\boldsymbol{\alpha}}$.
4: Compute $\widehat{\boldsymbol{v}}_i = \sqrt{\frac{\widehat{\alpha}_i}{\alpha_0(\alpha_0+1)}} \widehat{W}^\top \widehat{\boldsymbol{\mu}}_i$ for each topic $i$.
5: Recover linear classifier: $\widehat{\eta}_i \leftarrow \frac{\alpha_0+2}{2} \widehat{\boldsymbol{v}}_i^\top \widehat{M_y}(\widehat{W}, \widehat{W}) \widehat{\boldsymbol{v}}_i$.
6: **Output:** $\widehat{\boldsymbol{\eta}}$, $\boldsymbol{\alpha}$ and $\{\widehat{\boldsymbol{\mu}}_i\}_{i=1}^k$.

---

The sample complexity of Algorithm 1 depends on the sample complexity of the LDA inference algorithm $\mathcal{A}$. Although the LDA inference algorithm $\mathcal{A}$ is free to make any assumptions on the topic distribution matrix $O$, we comment that the linear independence of topic distribution vectors $\boldsymbol{\mu}$ is still required, because the power update trick (step 5 in Algorithm 1) is valid only $W\boldsymbol{\mu}_i$ are orthogonal vectors.

## Appendix C.2. Going beyond SVD

The proposed methods are based on spectral decomposition of observable moments and are provably correct. However, a major drawback of these methods is their assumption that the topic distributions $\boldsymbol{\mu}$ are linearly independent. Although standard LDA models assume this topic independence [BNJ03], in practice there is strong evidence that different topics might be related [BL12]. Therefore, it is important to consider possible extensions.

Recently, there has been much work on provable LDA inference algorithms that do not require the topic independence assumption [AGH+13, AGM12]. Instead of making the independence assumption, these methods assume a $p$-separability condition on the topic distribution matrix, that is, there exists an *anchor word* for each topic such that the anchor word only appears in this specific topic, and its probability (given that topic) is at least $p$.

Here we briefly describe an idea that might lead to an sLDA inference algorithm without assuming topic independence. First, let $O \in \mathbb{R}^{V \times k}$ be the topic distribution matrix defined previously and $H = (\boldsymbol{h}_1, \cdots, \boldsymbol{h}_N) \in \mathbb{R}^{k \times N}$ be the matrix of topic mixing vectors for each document. Suppose $Q = \mathbb{E}[\boldsymbol{x}_1 \boldsymbol{x}_2^\top]$ is the word co-occurrence frequency matrix and $Q_y = \mathbb{E}[y\boldsymbol{x}_1]$ is the co-occurrence frequency matrix between word and regression variable observations. It is clear that both $Q$ and $Q_y$ can be estimated from the training data, and we have

$$Q = ORO^\top, Q_y = OR\boldsymbol{\eta}, \tag{33}$$

where $R \triangleq \mathbb{E}[HH^\top]$.

Assuming the $p$-separability condition holds and using methods developed in [AGM12, AGH+13], we can consistently recover the $O$ and $R$. Now note that $Q_y = (OR) \cdot \boldsymbol{\eta}$. As a result, knowing both $OR$ and $Q_y$ we reduce the $\boldsymbol{\eta}$ inference problem to solving a linear equation systems. Future research could be done on (1) determining when the matrix $OR$ has full rank $r$ so that the linear equations can be solved, and (2) investigating the sample complexity problem for such an inference algorithm.

## Appendix C.3. From regression to classification

A natural extension to our result is to consider supervised LDA models for classification purposes. The simplest model is a logistic regression classification model, where we assume the response variable $y_d$ for each document $d$ is in $\{+1, -1\}$, and

$$\Pr[y_d = 1 | \boldsymbol{h}_d] = \frac{\exp(\boldsymbol{\eta}^\top \boldsymbol{h}_d)}{1 + \exp(\boldsymbol{\eta}^\top \boldsymbol{h}_d)},$$

where $\boldsymbol{\eta}$ is a linear classifier.

Though appears simple, such an extension incurs many fundamental problems. A major obstacle is the fact that the conditional expectation $\mathbb{E}[y|\boldsymbol{h}]$ is no longer linear in the topic mixing vector $\boldsymbol{h}$. As a result, we cannot even evaluate $\mathbb{E}[y]$ (or higher order tensors like $\mathbb{E}[y\boldsymbol{x}_1 \otimes \boldsymbol{x}_2]$) in closed forms. Further more, even if we have accurate estimation to the above moments, it is rather difficult to infer $\boldsymbol{\eta}$ back due to its non-linearity. To our knowledge, the classification problem remains unsolved in terms of spectral decomposition methods, and there is doubt whether a provably correct algorithm exists in such scenarios.