[Reviews · NeurIPS 2014]

Submitted by Assigned_Reviewer_4

This paper addresses the classical problem of unsupervised learning of latent topic model, with an extra variable called response, which can be a score. The main issue is that the topic model (and the embeddings deduced from it) may not help in learning this extra variable, as the response can be induced by phenomena that are orthogonal to the topics. The goal of the so-called supervised topic model learning is to drive the topic learning into a direction which makes it useful w.r.t. the prediction of this extra variable (by a regression).

The basic model considered is the Latent Dirichlet allocation (LDA) model. The parameters of such a model can be estimated with Max-Likelihood, sampling, or recently spectral methods (methods of moments). The spectral method is based on the consistent estimate of a finite number of observable statistical moments, from which it is possible to recover the parameters thanks to simple linear algebra operations (in the original paper) or more complex ones (like tensor decomposition).

Starting from previous work done on spectral learning of LDA, the paper shows how the same method can be used to learn the response variable, by adding an extra family of observable moments, and by showing how it is possible to recover the parameters of the regression via tensor decomposition.

Some experiments show that the algorithm is good w.r.t. other methods in parameter recover on a synthetic dataset. Conversely, the method does not perform well in real data experiments, with a max. likelihood objective. This is not surprising, as the non-spectral method usually focus on direct maximization of likelihood, whereas spectral does not. Also, the spectral method does not behave well in general with small samples (compared to the information needed for a good estimate)

The paper is well written, the mathematical part is solid.

The algorithm described in the paper, to my opinion, is more an extension of the spectral algorithm for LDA than a real new method. Its main interest is to show that the spectral method can be also applied to the sLDA model. More work needs to be done to make this method efficient in real data problems (as it is for spectral method in general).

(I think that when the authors say that « Theorem 1 is that it provides a necessary condition for a supervised LDA model to be identifiable », they mean in fact « sufficient », right?)

Pros:
- good adaptation of the spectral LDA learning algorithm
- good theoretical work
- theoretical bounds

Cons:
- more an extension of known methods than a new contribution
- not convincing in real data learning
Summary: The paper presents an extension of the spectral method for LDA to the supervised case. The theoretical part is solid. However, this contribution is mainly incremental and does not propose any new idea.

Submitted by Assigned_Reviewer_22

This paper presents a spectral algorithm for learning supervised Latent Dirichlet allocation (sLDA) models. This algorithm is an extension of the tensor method of Anandkumar et al. (2012). Specifically, the model and the learning algorithm are exactly the same except for a response variable y = eta^T h for each document where h is the topic distribution of the document and eta is an additional model parameter.

However, this extension is nontrivial. The paper derives an additional moment M_y which can be estimated from observable quantities to recover sLDA parameters. Furthermore, it introduces a trick of estimating eta with formerly estimated word distributions to dodge some problematic issues with orthogonalization.

The paper performs solid experiments on both synthetic and real datasets. The experimental results are not so surprising; (correct) spectral methods typically nail the learning problem when the data is actually generated by the model but fall a little behind when the data comes from the real world.

While novelty is not the strongest aspect of the paper, sLDA models are an important class of models, and the results of this paper will benefit other researchers pursuing similar directions.
Summary: This paper presents a nontrivial extension of the tensor method of Anandkumar et al. (2012) to estimate the parameters of supervised Latent Dirichlet allocation models and demonstrates the proposed method with solid experiments.

Submitted by Assigned_Reviewer_35

The authors propose a spectral method for estimating supervised LDA.
The empirical results are mostly good (at least for synthetic data),
although it is slightly worrying that
"Fig. 4 shows that in general Spectral-sLDA does not perform as good
as Gibbs sampling." The explanation offered " One possible reason is
that real-world datasets are not exact i.i.d. samples from an
underlying sLDA model." is not entirely convincing. Why would this harm the
performance of a spectral estimation scheme more than a Gibbs sampling one?
They are both estimating the same model, no?

Minor quibbles:

"does not perform as good as ..." should perhaps read ""does not perform as well as ..."

"This interesting observation shows that the Spectral-sLDA algorithm
can be used for model selection to avoid overfitting by using too many topics"
- that's one way to put a positive spin on a negative result!

"Spectral-sLDA is very easy to parallelize ..." Yes, but so is Gibbs sampling.

"A more serious fallacy is that when .." I think you mean "problem" rather than "fallacy"
"...reasonable performances." - should be "performance"
Summary: The authors propose a spectral method for estimating supervised LDA. The method appears sound
and the results are mostly reasonable, although I have some concerns about the results on the amazon data.
Author Feedback
Author rebuttal: We thank the reviewers for their helpful comments. We will improve the draft accordingly. Below, we address the common concern from Reviewers 2 and 3, as well as their detailed comments.

For the common concern of Reviewers 2 & 3 on the experiments on Amazon data:

We agree that when used alone, the spectral learning algorithm may fall a little behind than non-spectral ones, which is consistent with other results reported in the literature. As pointed out by Reviewer 3, this is because we are using a likelihood criterion which non-spectral methods directly optimize. However, we would like to point out that when Spectral-sLDA is combined with Gibbs sampling, the hybrid algorithm achieves superior performance in terms of both predicting accuracy and running time, as shown in Table 2. We believe this is a promising approach for learning sLDA models by first using an efficient, global but somewhat coarse algorithm to get a high-quality initialization and then use a less efficient local search method to further (directly) optimize the likelihood we are interested in. This also shows the practical effectiveness of our proposed algorithm.

To Reviewer 2:

Q1. Real-world datasets are not i.i.d. sampled from an underlying sLDA. Why would this harm a spectral estimation scheme more than a Gibbs sampling one?

Even if data are not exactly generated from an underlying sLDA distribution, the Gibbs sampling method could still work reasonably well by finding some estimates that are close to max-likelihood or MAP estimation from the sLDA parameter space. However, a spectral estimation scheme is based on consistent estimation of moments, so it could be more sensitive to the data model. From a theoretical point of view, when model assumptions do not hold we don't have theoretical guarantee on the estimation quality any more.

Q2. Our proposed algorithm is easy to parallelize, but so is Gibbs sampling.

When we said “parallelizable”, we meant that it is embarrassingly easy to parallelize our spectral learning method for the moment computation step, with very low communication and synchronization cost. However, for the Gibbs sampling for sLDA (or even LDA), parallelization is highly nontrivial. In fact, much work has been done on parallelizing it for LDA and the state-of-the art methods often rely on some smart asynchronization designs, e.g., [1,2,3]. We will include the discussions in the revision.

[1] A. Ahmed, M. Aly, J. Gonzalez, S. Narayanamurthy, & A. Smola. Scalable inference in latent variable models. In WSDM, 2012.
[2] D. Newman, A. Asuncion, P. Smyth, & M. Welling. Distributed algorithms for topic models. JMLR, (10):1801–1828, 2009;
[3] A. Smola & S. Narayanamurthy. An architecture for parallel topic models. VLDB, 3(1-2):703–710, 2010.

To Reviewer 3: more an extension of known methods than a new contribution:

We agree that our work is a natural extension of some known and provably correct methods. However, as agreed by Reviewer 1, we think this is an important and nontrivial investigation to generalize spectral methods to a new (and important) family of models, and it will benefit researchers pursuing similar directions. Both our theoretical and practical studies add new evidences and values on demonstrating the promise of spectral methods.